# Mechanistic Investigation of the Pyrolysis Temperature of Reed Wood Vinegar for Maximising the Antibacterial Activity of *Escherichia coli* and Its Inhibitory Activity

**DOI:** 10.3390/biology13110912

**Published:** 2024-11-08

**Authors:** Bing Bai, Meihui Wang, Zhongguo Zhang, Qingyun Guo, Jingjing Yao

**Affiliations:** 1Institute of Resources and Environment, Beijing Academy of Science and Technology, Beijing 100089, China; baibing_1029@163.com (B.B.); zn.zhang@163.com (Z.Z.); 2National Key Laboratory of Veterinary Public Health Security, College of Veterinary Medicine, China Agricultural University, Beijing 100193, China; wangmeihui718@163.com; 3Beijing Milu Ecological Research Center, Beijing 100076, China

**Keywords:** biofilm, natural antimicrobials, oxidative stress, pyrolysis, wood vinegar

## Abstract

Wood vinegar is a waste liquid recovered from the pyrolysis of wood to produce charcoal and a complex mixture. It is an environmentally friendly, high-quality product that can bring additional economic benefits to the plant. It is due to the diversity of its composition in agricultural practices that wood vinegar is a natural antimicrobial agent, but no study has yet investigated its antimicrobial mechanism. In this work, we evaluate the effects of wood vinegar on *Escherichia coli (E. coli)* in vitro, including inhibiting the growth curve of *E. coli*, disrupting the cellular morphology of *E. coli*, and crumpling *E. coli* cell membranes. In addition, the hypothesis that wood vinegar may inhibit biofilm formation in *E. coli* by suppressing the expression of *malE*, which in turn initiates a series of inhibitory effects, is also proposed in this study through transcriptome sequencing results. It provides a theoretical basis for the mechanism of wood vinegar as a natural antibacterial agent.

## 1. Introduction

As the main wetland species in China, reeds play a very important role in water purification, regional climate regulation, and wetland species protection [1]. With the massive growth of reeds, the withered reeds will decompose into water bodies after withering, causing eutrophication of wetland ecosystems and the production of black and odorous water bodies [2]. Currently, in order to dispose of excess reeds around wetlands, the main routes are through landfill [3], burning [4,5], or pyrolysis. Pyrolysis is a thermochemical process for the production of biochar as a sustainable biomass conversion and waste management method [6]. In order to address the excess vapour or fumes generated during pyrolysis [7], plant recovery and proper disposal of pyrolysis fluids is one way to turn the situation around, resulting in additional economic benefits, significant reductions in greenhouse gas emissions, and high-quality products with a variety of applications [8]. Wood vinegar is a by-product of recycling. Wood vinegar, also known as pyroligneous acid, is a complex organic mixture obtained by separating the vapours produced in biomass, such as wood or wood waste, in the pyrolysis equipment, which is then isolated from the air, cracked at high temperatures, and recovered by condensation [9]. The substance is initially separated directly by condensation reflux and is called crude wood vinegar [10]. As the crude liquid product contains tar and harmful substances, it needs to be refined to obtain refined wood vinegar for different purposes before it can be utilised or developed into a series of wood vinegar products to enter the market, thus bringing economic benefits to the factory.

Wood vinegar is readily available from widely distributed and low-cost sources and can be converted into products with a variety of uses, making its recycling essential for reducing air pollution and adding value to the carbonisation process by increasing the economic returns of the charcoal production chain [11]. Due to the versatility of its composition, wood vinegar can be applied as a plant growth regulator [12], insecticide [13], weed suppressant [14], etc., as an alternative to pesticides and fertilisers. In addition, wood vinegar can be used as a sterilant [15], soil conditioner [16], deodoriser [17], preservative [18], etc. It is widely used in the fields of agriculture and forestry, animal husbandry, environmental protection, industry, food processing and healthcare. It has been found that wood vinegar has an inhibitory effect on many kinds of pathogenic bacteria, and it is low cost with no residue and no pollution, so it has become a hotspot of biological control at present [19].

Many studies have demonstrated that wood vinegar possesses antimicrobial properties against pathogenic microorganisms as well as its natural, safe, and eco-friendly properties and has been recognized as a promising alternative to natural antimicrobial agents [20,21]. The results showed that wood vinegar inhibited bacteria such as *Escherichia coli *(*E. coli*)*, Pseudomonas aeruginosa *(*P. aeruginosa*)**, and *Staphylococcus aureus *(*S. aureus*)** [22]. In in vitro tests, researchers found that wood vinegar (Pyroligneous acid) also had high antibacterial activity at low concentrations [1.6% (*v*/*w*)] and against both *Salmonella enterica *(*S. enterica*)** and *Lactobacillus acidophilus *(*L. acidophilus*)** [23]. A study has shown that lychee wood vinegar has high antioxidant activity and antimicrobial activity, and the main components of lychee wood vinegar exhibited broad-spectrum inhibition of all bacterial strains, and most antibiotic-resistant strains were more susceptible to wood vinegar than non-antibiotic-resistant strains [24]. The bacteriostatic components of eucalyptus and bamboo wood vinegar may act as bacteriostatic by disrupting cell structure, decreasing cell membrane stability, and inhibiting protein synthesis and key gene expression [25]. Even so, there is still a gap in the research on the specific bacteriostatic mechanism of wood vinegar. In addition, pyrolysis temperature also has an effect on the bacteriostatic activity of wood vinegar. When the pyrolysis temperature increases, the acidic substances in eucalyptus wood vinegar increase while the phenolic substances decrease, and phenolic substances are considered to be the main substances exerting antimicrobial activity [26]. Interestingly, the concentration of phenol in wood vinegar does not increase with increasing temperature; it shows a different trend from 350 to 750 °C, first increasing and then decreasing maximally at 550 °C [27]. Therefore, since wood vinegar is a compound with a complex composition, its antimicrobial activity is not proportional to the pyrolysis temperature. The antimicrobial activity of wood vinegar in *Eucommia ulmoides* Olivers branches at different pyrolysis temperatures was studied by Hou et al. [28]. The antimicrobial activity of wood vinegar at 300–330 °C was noted to be better against *Enterococcus aerogenes, E. coli,* and *Bacillus subtilis*, whereas the antimicrobial activity of wood vinegar at 240–270 °C was better against *S*. *aureus* and *B*. *cereus*. In addition, the temperatures showing good antimicrobial activity against different microorganisms include 240–270 °C, 270–300 °C, 300–330 °C, and 450–480 °C compared to other temperatures, and experiments have demonstrated that the optimal pyrolysis temperatures are mainly in the mid-temperature range [28]. Accordingly, the better antimicrobial activity of wood vinegar at pyrolysis temperature is still a question worth exploring.

Therefore, although reeds are a dominant plant in many wetlands, their uncontrolled growth has seriously affected the growth of other vegetation and caused eutrophication of wetland ecosystems. In order to achieve the goal of protecting wetlands and utilize the large-scale growth of wetland reeds, this study determines the optimal pyrolysis temperature for maximizing the antibacterial activity of reed wood vinegar against *E. coli* and elucidating its underlying mechanism. The study explored the antimicrobial potential of reed wood vinegar at different pyrolysis temperatures using micro broth dilution, agar plate diffusion, Quantitative Polymerase Chain Reaction (qPCR), crystal violet assay, and antioxidant level assay to select the best wood vinegar for a bacteriostatic effect. Through RNA sequencing (RNA-Seq), we focused on its inhibitory effect on the biofilm of *E. coli* and its mechanism of inhibition, with the aim of using charcoal effectively, avoiding environmental pollution caused by the direct burning of wood and becoming a feasible solution to the problem of microbial resistance to increase the eco-efficiency and economic efficiency of enterprises.

## 2. Materials and Methods

### 2.1. Determination of Strain Origin and Its Culture Conditions

The *E. coli* used for experiments in this study was obtained by collecting, isolating, and culturing from the soil around the faeces of Milu in the Beijing Milu Park. The experimental procedure was as follows: the collected soil was mixed with Phosphate-Buffered Saline (PBS) (Solarbio, Beijing, China) in equal amounts, and the mixture was streaked on a plate of MacConkey solid medium (Luqiao, Beijing, China) with an inoculating loop, then incubated in a bacterial incubator at 37 °C for 24 h. After that, a single colony was picked and cultured in Luria–Bertani (LB) liquid medium (Luqiao, Beijing, China) overnight at 37 °C.

The isolated *E. coli* was mixed with LB medium to obtain four concentration gradients of 1.0 × 10^5^, 1.0 × 10^4^, 1.0 × 10^3^, and 1.0 × 10^2^ CFUs/mL, respectively, and placed in a 37 °C bacterial incubator to take samples at 0 h, 1 h, 2 h, 3 h, 4 h, 5 h, 6 h, 7 h, and 8 h. The OD_600_ of *E. coli* was measured by a spectrophotometer (Jinghua, Shanghai, China), and the growth curve of *E. coli* was plotted according to the relationship between time and OD_600_.

### 2.2. Identification of the ST Type of E. coli

According to the manufacturer’s instructions, *E. coli* whole DNA was extracted using the TIANamp Bacteria DNA Kit (TianGen, Beijing, China), and samples were stored at −20 °C. Referring to Multilocus Sequence Typing (MLST) Allelic Profiles and Sequences [29], primers were designed and PCR-amplified in a SimpliAmp PCR thermal cycler (Thermo Fisher Scientific, Waltham, MA, USA) (Table 1), and primers were synthesized by Shanghai Sangon Biotech Co. The PCR system was 25 μL, including 8.5 μL of double-distilled water, 1 μL of forward primer, 1 μL of reverse primer, 12.5 μL of 2 × Taq Master Mix (Dye Plus) (Vazyme, Nanjing, China), and 2 μL of genomic template DNA of *E. coli*. PCR conditions consisted of DNA pre-denaturation at 95 °C for 3 min, followed by 35 cycle cycles including the following steps: denaturation of DNA at 95 °C for 30 s, annealing for 30 s, elongation of DNA at 72 °C for 40 s, and finally extension at 72 °C for 10 min. PCR products were sequenced by Shanghai Sangon Biotech Co. (Sangon, Shanghai, China). The target fragment sequences were homology searched by the pubmlst program (https://pubmlst.org/organisms/escherichia-spp, accessed on 26 February 2024) in National Council for Biotechnology (NCBI) and thus obtained by *E. coli* Sequence Type (ST) typing.

### 2.3. Preparation of Reed Wood Vinegar

In this study, naturally dried reeds were used, which were crushed and then made into cylindrical pellets with diameters of about 0.5 cm and lengths of about 3 cm by a compression granulator. The reed pellets were evenly spread in a 304s stainless steel basin and then placed into a high-temperature pyrolysis carbonisation furnace (this was a customised furnace) for pyrolysis. The heating rate of the furnace was set at 10 °C/min, nitrogen was continuously fed at a flow rate of 3 mL/min, and the pyrolysis reaction was maintained for 4 h after reaching the target temperatures (set temperatures of 300 °C, 500 °C, and 700 °C, respectively). The flue gas generated in the pyrolysis process produced condensed liquid when passing through the discharge pipe, the flue gas condensed liquid in the pyrolysis process was collected to obtain the crude reed wood vinegar at the corresponding temperature, which was collected and sealed in brown bottles for storage. After that, the thicker impurities in the crude wood vinegar were filtered out by qualitative filter paper. A small amount of activated carbon was added to the liquid obtained (the added amount was about one tenth of the volume of the wood vinegar), and then sealed and put into a constant-temperature oscillation incubator to oscillate for 12 h. The supernatant was aspirated and filtered through a 0.45 μm filter membrane to obtain the refined wood vinegar, which was sealed in a brown bottle and stored at a low temperature. Eventually, 300 °C wood vinegar (LWV), 500 °C wood vinegar (MWV), and 700 °C wood vinegar (HWV) were obtained.

### 2.4. Determination of the Minimum Inhibitory Concentration (MIC) of Reed Wood Vinegar

First, the susceptibility of *E. coli* obtained from the present experimental isolates to reed wood vinegar was evaluated in vitro to determine the MIC. The MIC determination was performed according to the Clinical and Laboratory Standards Institute [30] standards using the micro broth dilution method. Bacteria were monoclonalized in BHI broth medium (Luqiao, Beijing, China) and incubated in a bacterial incubator (Boxun, Shanghai, China) at 37 °C for 24 h. Subsequently, the process of microbial inoculum preparation was performed. The bacterial OD600 was diluted to 0.5 (approximately 1.0 × 10^8^ CFUs/mL) using a spectrophotometer (Jinghua, Shanghai, China) and was prepared by diluting it 100-fold with BHI broth medium. Then, the reed wood vinegar at different pyrolysis temperatures (LWV, MWV and HWV) was diluted by 2 times to the 10th well in a 96-well plate, with 100 μL of Brain–Heart Infusion Broth (BHI) medium added. In addition, the 11th column was a negative control containing only BHI medium, and the 12th column was a positive control containing the bacterial solution to be tested, respectively. After dilution, 100 µL of the corresponding microbial inoculum was added to each microtiter well. Immediately after inoculation, the microtiter plates were read at 630 nm using an enzyme-linked immunoassay analyser (Thermo Fisher Scientific, MA, USA) to record the absorbance at the time of inoculation (0 h). Then, the microtiter plates were incubated in a bacterial incubator at 37 °C for 16–18 h. Afterwards, a visual inspection was performed to determine the MIC, with three microtiter wells being completely translucent as a decision criterion for determining this parameter. The first visual inspection was compared with the absorbance read on a spectrophotometer after 16–18 h to provide quantitative data on the activity of *E. coli* under the influence of reed wood vinegar at different pyrolysis temperatures and each wood vinegar at different concentrations. All tests to determine the MIC were performed in triplicate with three replicates each.

### 2.5. Effect of Reed Wood Vinegar on the Growth Curve of E. coli

This experiment was conducted based on the method of Lan et al. [31]. The growth curve method was used to determine the effect of reed wood vinegar with different pyrolysis temperatures on the growth curve of *E. coli*. Briefly, *E. coli* cultured to the logarithmic phase (final concentration of 1 × 10^6^ CFUs/mL) and different concentrations (1/2 MIC and 1 MIC) of LWV, MWV, and HWV were added to centrifuge tubes in a final volume of 40 mL. The same volume of bacterial suspension without reed wood vinegar was used as a control to eliminate the effect of the samples to be tested on the absorbance values. The centrifuge tubes were incubated in an incubator at 37 °C. Samples were taken at 0, 2, 4, 6, 8, 10, 12, 14, 16, 18, 20, 22, and 24 h. The absorbance values at 600 nm were determined by a spectrophotometer, the incubation time was taken as the horizontal coordinate, the measured OD values were taken as the vertical coordinate, and the 24 h growth curve was plotted. Each treatment was repeated three times.

### 2.6. Reed Wood Vinegar Antimicrobial Activity Assay

Based on Song et al. [32] with modifications, the effect of reed wood vinegar on the size of the circle of inhibition of *E. coli* was determined by the diffusion method using filter paper agar plates. The specific experiments were as follows: *E. coli* cultured to the logarithmic stage (OD_600_ = 1) was taken, dipped in a bacterial solution with a sterile cotton swab, and coated on the whole surface of the solid plate of tryptic soy agar solid medium (Qingdao hopebio, China) so that the whole plate was coated evenly. Finally, the edges of the plate around the edges of the plate were coated with a cotton swab. We took a sterile, drug-sensitive paper sheet on a clean plate and added 10 μL and 15 μL of LWV, MWV, and HWV. Sterile saline was used as a negative control. The plate coated with bacterial solution was dried at room temperature for 3–5 min; then, the drug-sensitive paper sheet with sterile tweezers was taken and stuck onto the surface of the plate. The paper sheet was pressed gently with the tweezers to make it stick flat. The plate was incubated at 37 °C for 18–24 h, and then the diameter of the inhibition circle was measured with vernier calipers. The presence or absence of an inhibition circle and the diameter size of the inhibition circle were taken as the indicators of the inhibitory effect of reed wood vinegar on the growth of *E. coli* at different pyrolysis temperatures.

### 2.7. qPCR for Detection of Gene Expression

Cell membrane genes in bacteria were selected for real-time quantitative PCR analysis [33]. First, *E. coli* was cultured to the logarithmic growth stage in LB medium at 37 °C and diluted to OD_600_ of 0.6. Then, 1/4 MIC or 1/2 MIC of LWV, MWV, and HWV were added, and the samples were co-cultured for 6 h. The samples were collected and transferred to centrifuge tubes, which were washed twice with PBS buffer and centrifuged for 5 min at 6000 rpm. Total RNA was extracted using the RNAprep Pure Cell/Bacteria Kit (TianGen, China) to extract total bacterial RNA. Next, RNA was reverse-transcribed using the HiScript III 1st Strand cDNA Synthesis Kit (+gDNA wiper) (Vazyme, Nanjing, China) into cDNA. cDNA was diluted 10-fold and subjected to polymerase chain reaction on an ABI SimpliAmp PCR instrument (Applied Biosystems, USA). qPCR was performed with ChamQ SYBR qPCR Master Mix (High ROX Premixed) (Vazyme, Nanjing, China). Primers for qPCR were obtained by querying the gene database at NCBI (https://www.ncbi.nlm.nih.gov/, accessed on 7 March 2024) and were synthesized by Bioengineering Biotechnology Co. (Bioengineering Biotechnology, Shanghai, China) (Table 2). The relative expression level of genes was represented by 2^−ΔΔCt^, and 16sRNA was used as a reference gene. 2^−ΔΔCt^ was calculated as follows:$${Fold\;change=2}^{-[{Ct}_{e1}-{Ct}_{e2}]-[{Ct}_{c1}-{Ct}_{c2}]}$$

Ct_e1_ denotes the expression of the target gene in the experimental group, Ct_e2_ denotes the expression of the internal reference gene in the experimental group, Ct_c1_ denotes the expression of the target gene in the control group, and Ct_c2_ denotes the expression of the internal reference gene in the control group.

### 2.8. Analysis of Biofilm Formation of E. coli by Reed Wood Vinegar Solution

We characterized the ability of *E. coli* biofilm formation using the crystal violet method [31]. *E. coli* cultures (OD_600_ = 0.1) were diluted at 1:10, 200 μL/well of the dilution solution was added into 96-well enzyme-labelled plates, and 3 wells of each group were replicated in parallel and incubated in an incubator at 37 °C for 24 h. At the end of the incubation period, the culture medium was discarded, and the plates were rinsed gently with PBS 3 times. The 96-well plates were supplemented with LWV, MWV, and HWV of 1/4 MIC and 1/2 MIC for 24 h, then washed with PBS after the removal of planktonic bacteria and fixed with 100 μL of anhydrous methanol for 15 min after drying. After 24 h of treatment with HWV to remove planktonic bacteria, the plates were washed with PBS and fixed with 100 μL of anhydrous methanol for 15 min. After drying, the plates were stained with 200 μL of 1% crystal violet (Solarbio, Beijing, China) for 30 min, washed with PBS, and dissolved with 200 μL of 30% glacial acetic acid for 30 min. The absorbance value at a wavelength of 570 nm was measured using an enzyme-linked immunoassay analyser (Thermo Fisher Scientific, MA, USA). Meanwhile, the same volume of LB culture solution was set up as a blank control well, and the test was repeated three times to characterize the effect of reed wood vinegar distilled at different temperatures on the ability of *E. coli* biofilm formation.

### 2.9. Scanning Electron Microscopy (SEM) and Transmission Electron Microscopy (TEM) Inspection

The *E. coli* suspension was incubated overnight at 37 °C in LB medium, and the bacterial concentration was adjusted to 1 × 10^8^ CFUs/mL, to which 1/4 MIC and 1/2 MIC of LWV, MWV, and HWV were added. The bacterial suspension was used as a control and then incubated overnight at 37 °C in a bacteria culture incubator. At the end of the incubation, the samples were centrifuged with an Eppendorf centrifuge at 3000 rmp for 10 min, and the medium was discarded and gently rinsed with PBS, then PBS was discarded. In addition, 2.5% glutaraldehyde solution (Leagene Biotechnology, Beijing, China) was added to the samples and the bacteria were blown open and suspended in the fixative for 2 h at room temperature, and then transferred to 4 °C for 24 h to fix the samples.

For SEM, samples were rinsed three times with PBS; fixed with 1% osmium tetroxide for 1 h; and subsequently dehydrated at room temperature using concentrations of 50%, 70%, 80%, 90%, and 100% ethanol in that order. The samples were dried in a desiccator for 12 h, adhered to a carbon tape, and sprayed with gold on the carbon tape. Finally, the samples were observed under a field emission scanning electron microscope, SU8100 (HITACHI, Japan).

The steps of the TEM assay were as follows: the samples were fixed by 1% osmium fixative and rinsed with PBS; dehydrated by 50% ethanol, 70% ethanol, 90% ethanol, 90% ethanol:90% acetone (1:1), 90% acetone, and 100% acetone; embedded in EPON812 embedding agent at 37 °C for 2–3 h; and fixed in an oven. Afterwards, the samples in the embedding agent were cut into sections with thicknesses of 70 nm using a Leica EM UC7 ultrathin microtome (Leica, Wetzlar, Germany), the sections were stained with 2% uranyl acetate–lead citrate, and the samples were visualized and photographed by observation under a transmission electron microscope, HT7800 (80 KV) (HITACHI, Tokyo, Japan).

### 2.10. E. coli Malondialdehyde (MDA) and Nitric Oxide (NO) Detection

MDA and NO levels in *E. coli* were used to assess the effect of wood vinegar distilled at different temperatures on the antioxidant levels of *E. coli*. Specifically, *E. coli* cultures grown in logarithmic phase were incubated with 1/4 MIC and 1/2 MIC of LWV, MWV, and HWV for 24 h. *E. coli* were collected and bacterial proteins were extracted by a mixture of RIPA Lysis Buffer (Servicebio, Wuhan, China) and protease lysate (New Cell & Molecular Biotech Co., Suzhou, China), and protein concentration was measured using a Bicinchoninic Acid Assay (BCA) kit (Beyotime, Shanghai, China). Regarding the MDA and NO (Beyotime, China) levels in *E. coli*, the TBA reagent was used to determine the MDA level in *E. coli* and the Griess reagent was used to determine the NO level in *E. coli* according to the instructions of the reagent vendors. Intercept a and slope b in the standard curve were calculated from the OD540 of the standard.
$$b=\frac{n\sum (x_{standard}y_{standard})-\sum (x_{standard}y_{standard})}{n\sum (x_{standard}^{2})-{\left(\sum x_{standard}\right)}^{2}}\;a=\frac{\sum y_{standard}}{n}-b\frac{\sum x_{standard}}{n}$$

x_standard_ represents the concentration of the standard, y_standard_ represents the OD540of the standard, a represents the intercept, b represents the slope, and n represents the number of test samples.

Then, the OD_540_ of the sample was brought into the following equation to obtain the concentration of MDA or NaNO_2_.
$$c_{MDA/{NaNO}_{2}}=\frac{{OD}_{540}-a}{b}$$

c stands for concentration.

### 2.11. Detection of Bacterial Adenosine Triphosphate (ATP) Content

*E. coli* was incubated at 37 °C until the logarithmic growth period. The samples were centrifuged at 5000 rpm for 8 min at room temperature; the supernatant was discarded, washed three times with PBS, and resuspended; and the bacterial suspension was adjusted to an OD_600_ of approximately 0.5. *E. coli;* treated with 1/2 MIC or 1/4 MIC of LWV, MWV, and HWV; and incubated at 37 °C for 8 h before centrifugation at a low temperature. Bacterial precipitates were added with 1 mg lysozyme (Beyotime, China), vortexed thoroughly, and centrifuged at 4 °C. Then, 20 μL of supernatant was added to the configured ATP assay working solution and mixed, and chemiluminescence values were measured with an INFINITE M PLEX multifunctional enzyme labeller (Tecan, Männedorf, Switzerland). The assay was performed using the Enhanced ATP Assay Kit (Beyotime, Shanghai, China) according to the vendor’s technical manual. Three biological replicates were set for each treatment.

### 2.12. Transcriptome Sequencing Analysis

Transcriptome sequencing was performed by Biomarker Biotechnology Co. MWV, with 1/2 MIC were added to the logarithmically grown *E. coli* culture medium and incubated for 24 h. First, 10 mL of the bacterial fluid with OD_600_ = 1 was collected. The precipitate was centrifuged at 6000 rpm for 5 min and then transferred to a 1.5 mL sterile freezing tube after discarding the medium, rinsing gently with PBS, and then discarding the PBS. Then, the bacterial precipitate was quickly put into liquid nitrogen for quick freezing for 5–10 min, followed by transferring to −80 °C. Extraction of total RNA from *E*. *coli* by trizol (Vazyme, Nanjing, China) was performed. After RNA purification, the samples were reverse-transcribed into cDNA under the action of reverse transcriptase, and then the cDNA libraries were sequenced using the NovaSeqXPlus (Version 2.0) (Illumina, CA, USA) and DNBSEQ-T7 (Version 3.0) (MGI Tech Co., Ltd., Shenzhen, China) platforms to remove the junction sequences and low-quality reads, as well as to obtain high-quality clean data. Sequence assembly of clean data was performed to obtain the Unigene library of the species. The statistical power calculated in RNASeqPower for this experimental design was 0.85. FDR < 0.05 and fold change ≥ 2 were used as the screening criteria to identify the differential expressed genes (DEGs) of LWV, MWV, and HWV affecting the growth of *E. coli* in 1/4 MIC and 1/2 MIC. The DEGs were plotted with log2 (fold change) as the horizontal coordinate and lg (false discovery rate) as the vertical coordinate DEGs volcano map. The screened DEGs were subjected to Gene Ontology (GO) and Kyoto Encyclopedia of Genes and Genomes (KEGG) enrichment analyses, and the genes and signalling pathways with *p* < 0.05 were screened, which were considered to be significantly enriched. Next, GO functional enrichment analysis was used to clarify the functional attributes of the DEGs and the gene products, and KEGG systematic analyses were performed to identify the major biometabolic pathways and their functions in which the screened DEGs were involved. Transcriptome analyses were conducted on the BMKcloud platform (https://www.biocloud.net/, accessed on 1 September 2024).

### 2.13. Statistical Analysis

In this experiment, all experimental results were analysed using SPSS 20.0 and presented in the form of mean ± standard error of mean. Figures were generated by GraphPad Prism 8.1. The circle of inhibition, qPCR, biofilm assay, and oxidative stress assay were subjected to one-way ANOVA for comparison between multiple groups. In addition, MIC, growth curve, and RNA-Seq results were compared between two groups using the independent *t*-test: * *p* < 0.05; ** *p* < 0.01; *** *p* < 0.001.

## 3. Results

### 3.1. Investigation of E. coli Culture, Types and Growth Conditions

In order to obtain *E. coli* in the environment, we collected soil around Milu faeces in the Beijing Milu Park to explore the growth conditions of *E. coli* in the environment. At first, we inoculated the soil samples in MacConkey solid medium at 37 °C for 24 h, after which isolated red *E. coli* colonies were observed (Appendix A). Then, single colonies were picked and incubated in LB liquid medium at 37 °C for 8 h. OD_600_ of *E. coli* was detected every hour and plotted as in Appendix A for *E. coli* incubation time and OD_600_. The results showed that *E. coli* entered the logarithmic growth period after 2 h of incubation at 37 °C and entered the stabilization period after 4 h. Therefore, we determined the duration of the logarithmic growth period of *E. coli* to be 2–4 h, thus facilitating the in-depth study of the following inhibition experiments. In addition, in order to clarify the type of *E. coli*, we designed primers based on the published *E. coli* housekeeping genes and sequenced them after amplification by PCR, and the results were compared in Pubmlst to determine the ST type of the *E. coli* used for this experiment as 101 (Appendix A).

### 3.2. Inhibitory Effect of Reed Wood Vinegar on the Growth of E. coli

To determine the inhibitory effect of reed wood vinegar with different pyrolysis temperatures on the growth of *E. coli*, we foremost obtained reed wood vinegar with different pyrolysis temperatures (Appendix A). Subsequently, we detected the MIC of reed wood vinegar with different pyrolysis temperatures against *E. coli* using the micro broth dilution method. From Figure 1A, it can be found that all the three types of reed wood vinegar with different pyrolysis temperatures were effective antimicrobial agents to inhibit the growth of *E. coli*. The results showed that the MIC values of both LWV and MWV were 0.1563 mg/mL, while the MIC of HWV was 1.25 mg/mL; therefore, LWV and MWV had more inhibitory ability.

Next, we calculated the inhibition curves of the three WVs at different concentrations (1/2 MIC and 1 MIC) against *E. coli* (Figure 1B–D). As can be seen from Figure 1B–D, LWV, MWV, and HWV all had significant inhibitory effects on the growth of *E. coli* compared to the control, and the onset of the logarithmic growth period of *E. coli* was delayed with the addition of reed wood vinegar. The most significant inhibitory effect of MWV was observed, and the curve showed that the bacterial concentration in the medium began to increase only after 10 h of contact between MWV and *E. coli*. In addition, we also evaluated the bacterial inhibition of reed wood vinegar with different pyrolysis temperatures at the same volume using the filter paper agar plate diffusion method. As shown in Figure 2A,B, we found that 10 μL and 15 μL of LWV and MWV had significant inhibitory effects on *E. coli* compared with the saline group by measuring the radius of the circle of inhibition, and 15 μL of reed wood vinegar significantly increased the radius of inhibition. Finally, this study also found that ATP expression was significantly reduced in *E. coli* with reed wood vinegar addition compared to the control group (Figure 2C). It is noteworthy that, among the six groups with reed wood vinegar added, the lowest ATP content was found in the 1/2MIC MWV-treated *E. coli*, and this result is also consistent with the MIC, which confirmed side by side that the best inhibitory effect of reed wood vinegar at 500 °C pyrolysis temperature significantly affected the ATP synthesis pathway, leading to the inhibition of *E. coli*’s normal physiological activities and to its death. Consequently, choosing a reasonable dose and program of reed wood vinegar can effectively inhibit the growth of *E. coli*.

### 3.3. Effect of Reed Wood Vinegar Solution on E. coli Biofilm

It has been shown that biocides primarily target the cytoplasmic membrane and interact with multiple targets on the bacteria, and that the number of bacteria affected and the severity of the damage result in an irreversible bactericidal effect or a reversible bacteriostatic effect. To begin with, we determined the effects of different concentrations and different pyrolysis temperatures of reed wood vinegar on the biofilm formation of *E. coli* through a crystal violet experiment. According to the crystal violet staining analysis, *E. coli* can form a biofilm in vitro and stain the nucleus of the cell into a dark purple colour with the crystal violet solution. As shown in Figure 3A,B, the biofilm formation of *E. coli* was significantly decreased by the addition of reed wood vinegar compared with the control, and the inhibition of biofilm showed a concentration-dependent effect. Particularly, 1/2MIC LWV or 1/2MIC HWV inhibited the biofilm formation of *E. coli* more significantly than 1/4MIC LWV or 1/4MIC HWV (*p* < 0.01), and 1/2MIC MWV significantly reduced the biofilm production of *E. coli* compared to 1/4MIC MWV (*p* < 0.01). Furthermore, among the six reed wood vinegar levels tested, 1/2MIC MWV exhibited the highest biofilm inhibition in *E. coli,* with the lowest OD of 0.183. This result was at least proven by the qPCR results, in which the expression of biofilm-associated genes at the mRNA level, such as motA, motB, and flgB, was significantly inhibited in the reed wood vinegar-added *E. coli*. Likewise, the expressions of ydiM, nanC, and mdtO related to cell membrane transporter proteins, as well as lsrA related to cellular interactions, were also significantly suppressed (Figure 3C–J). Altogether, these results suggest that the cell membrane may be a potential target for reed wood vinegar.

### 3.4. Effect of Reed Wood Vinegar on E. coli Morphology

In the following step, in order to more intuitively reflect how the morphology of *E. coli* changed after exposure to different concentrations of reed wood vinegar, we carried out microscopic observations by SEM and TEM. The micrographs obtained by SEM showed that the *E. coli* organisms in the control group were structurally intact, with a full appearance. The organisms were nearly ellipsoidal, the surface was smooth, and the refractive property was good, while the number of bacteria decreased after adding 1/4MIC reed wood vinegar. The surfaces of a large number of bacteria were rough, the refractive property decreased, and the sizes of individual bacteria decreased. When 1/2MIC of reed wood vinegar was added, the number of organisms decreased significantly, the volume was obviously reduced, and the surface was wrinkled and became extremely rough. Some of the organisms were irregular in morphology, with serious depression and distortion, and the morphology was seriously damaged (Figure 4). The TEM results showed that the membranes of *E. coli* cultures were unbroken, and the inner and outer membranes were clearly discernible, with normal morphology. However, after the addition of reed wood vinegar, it was observed that the cell membranes were severely disrupted, the membranes were visibly torn, the cytoplasmic content was reduced, and the bacteria were unable to maintain their rod shapes. In some cases, cell walls/membranes without cytoplasm were observed, implying lysis and death of the bacteria. Combining the above results, MWV showed the most significant disruption of *E. coli* morphology among all reed wood vinegar treatment groups (Figure 5).

### 3.5. Oxidative Damage to E. coli Caused by Reed Wood Vinegar

Recent studies suggest that killing of bacteria by various antimicrobial classes may involve reactive oxygen species (ROS) causing oxidative damage, just as a common self-destructive reaction occurs in *E. coli* exposed to antibiotics. For this purpose, we assessed the ability of reed wood vinegar to induce oxidative stress in *E. coli* by examining MDA and NO levels. Appendix A show that the MDA level in *E. coli* was significantly increased while the NO level was significantly decreased by the addition of either LWV, MWV, or HWV compared to the control. Simultaneously, the MDA of *E. coli* at a 1/2 MIC reed wood vinegar concentration was higher, while the NO content was lower than that of 1/4 MIC reed wood vinegar under the same pyrolysis temperature treatment. Therefore, the higher concentration of reed wood vinegar promotes the inhibition of lipid oxidation and nitric oxide production in *E. coli*, which causes oxidative stress.

### 3.6. Effect of Treatment with MWV on the Transcriptome of E. coli

In order to reveal the molecular mechanism of the inhibitory activity of *E. coli* by reed wood vinegar solution, we analysed the transcriptome results. From the RNA-seq results, we found that the number of genes shared between the control and MWV groups was 3292, the number of genes uniquely expressed in the control group was 284, and the number of genes uniquely expressed in the MWV group was 55 (Appendix A). From Appendix A, it can be found that there is high similarity among the control group, and at the same time, there is significant clustering among the MWV group, while there is a significant difference in the distribution between the control group and the MWV group. The correlation analysis result confirms the results of Principal Component Analysis (PCA), that is, the correlation coefficient of the samples in the control group and the treatment group is close to 1, indicating that the expression of each sample in the group is similar and well correlated (Appendix A). Cluster of Orthologous Groups (COG) classification statistics of the gene sets showed that the genes were mainly concentrated in (E) amino acid transport and metabolism, (G) carbohydrate transport and metabolism, (K) transcription, (M) cell wall/membrane/envelope biogenesis, and (X) mobilome: prophages, transposons (Appendix A). Likewise, we also found genes mainly concentrated in the metabolic process and cell membrane and catalytic activity in the statistical graph of GO analysis (Figure 6A). Moreover, we performed lineage analysis of the GO pathway for structural relationships, and from Appendix A, we were able to clearly find that the assembly of ribosomal large and small subunits by MWV is the underlying cause of the effect on the structure of *E. coli*, which further affects the protein–RNA complex assembly. In addition, the assembly of large and small ribosomal subunits affects bacterial membrane function and cellular composition. KEGG pathway enrichment aggregation revealed significant differences in the expression of genes primarily involved in ribosome, oxidative phosphorylation, and cationic antimicrobial peptide (CAMP) resistance (Figure 6B).

### 3.7. MWV Inhibits E. coli via malE

In the next step, the genomic results of *E. coli* were analysed in order to find the key genes for MWV inhibition of *E. coli*. The genome of *E. coli* was clustered into 10 groups, of which the third group clustered the highest number of genes, with 834 DEGs (Appendix A). Expression counts of the 834 DEGs in the third group showed that *E. coli* gene expression was significantly reduced in the MWV group compared to the control group (Appendix A). We counted the expression of the top 20 genes from the 834 genes with the most obvious differences, and the results showed that *repFIB*,* phoH*,* zinT*,* lrhA*,* yqiJ*,* novel0265*,* yejG*,* hspQ*,* novel0116*,* bssS*,* yebG*,* yqgA*,* uspF*,* aceA*,* fau*,* sopA*,* bssR*,* caiA*,** and *ycdT* were all significantly reduced in expression in the MVV group (Figure 7A). The functions played by the above genes in *E. coli* were queried by gene annotation as follows: *yqiJ, yqgA, bssR, bssS*, and *ycdT* were associated with cell membrane components; *phoH*,* uspF*,* fau*,** and *caiA* were associated with oxidative stress; and *repFIB, lrhA, hspQ*, and *yebG* were associated with DNA binding and repair (Appendix A). By subjecting the 834 gene-edited proteins to protein interaction network analysis, we found that *malE* proteins had the richest interactions with each protein and linked cell membrane-associated proteins (degP, ompF, nanC, etc.), ATPase-associated proteins (modC, ytrF, lsrB, etc.), and bacterial stress-associated proteins (groS, groL, etc.) affecting *E. coli*’s multiple biological processes (Figure 7B). Examination of the transcriptome results showed that the *malE* gene was significantly reduced in *E. coli* (*p* < 0.01). Therefore, we believe that *malE* proteins play a key role in the inhibition of *E. coli* by MWV, and further exploration of *malE* will follow.

## 4. Discussion

In this research, we investigated the bacteriostatic effect of reed wood vinegar, a pyrolysis product of reeds, at different pyrolysis temperatures and examined its mechanism of inhibition against *E. coli* extracted from the animal environment. The results showed that the reed wood vinegar obtained with a 500 °C pyrolysis temperature had the optimal antibacterial effect on *E. coli*, which could inhibit the growth vitality of *E. coli*, destroy the morphology and biofilm integrity of *E. coli*, and promote the oxidative stress of *E. coli*, which affected the normal growth and metabolism of the bacteria. The transcriptome results showed that the inhibitory mechanism of reed wood vinegar at a 500 °C pyrolysis temperature was related to the inhibition of DNA and ribosome synthesis, the inhibition of cell membrane production and activity, and the promotion of oxidative stress in *E. coli*, which affected bacterial division and reproduction and inhibited bacterial growth. The experimental results also indicate that the reed wood vinegar antibacterial agent is more suitable for preparation at 500 °C.

In production, biomass pyrolysis produces a large number of volatiles, the recycling of which is in line with the concept of sustainable development [34]. The water-soluble organic matter produced from pyrolysis volatiles is often referred to as reed wood vinegar and is widely used in agriculture as a high value-added product of biomass pyrolysis. Reed wood vinegar is a complex mixture with a rich composition, with acids and phenols as the main components, and also containing organic substances such as ketones, aldehydes, alcohols, esters, ethers, and furans [35]. Recently, in order to avoid environmental pollution and residue problems caused by synthetic pesticides, researchers have studied the antimicrobial properties of reed wood vinegar quite intensively, mainly focusing on the effects of different pyrolysis parameters. The common pyrolysis temperatures of reed wood vinegar are between 0 °C and 700 °C [36], and the pyrolysis temperatures selected in this study were 300 °C, 500 °C, and 700 °C, respectively. As the pyrolysis temperature increases, the concentration of reed wood vinegar in solution increases; the phenolic content increases; the content of alcohols, furans, acids, and ketones decreases; and the solution becomes clearer due to increased solubility as a result of increased polarity [12]. It is noteworthy that, due to the high pyrolysis temperature (>400 °C), reed wood vinegar with low-molecular-weight substances can easily enter the cell wall and thus inhibit bacterial growth [37]. In addition, Oramahi et al. have shown that the antimicrobial activity of wood vinegar produced by pyrolysis against *Reticulitermes speratus* and *Coptotermes formosanus* at 450 °C was higher than the antimicrobial activity at 400 °C and 350 °C [38]. Nevertheless, a study by Hou et al. categorized *Eucommia ulmoides* Olivers reed wood vinegar into low (90–210 °C), medium (210–360 °C), and high (360–510 °C) temperatures based on the preparation temperature. In exploring the differences in their antimicrobial effects, it was found that the medium-temperature wood vinegar was the most effective in terms of its antibacterial activity against *E. coli*, whereas the medium- and high-temperature wood vinegar were more effective against *Bacillus subtilis* [28]. Differently, in this study, although reed wood vinegar with different pyrolysis temperatures was able to delay the onset of the logarithmic growth period of *E. coli*, the reed wood vinegar obtained at a 500 °C pyrolysis temperature showed the best inhibitory effect on *E. coli*, which may have been due to the fact that the antimicrobial activity of the reed wood vinegar was closely related to the characteristics of the tested bacteria. Therefore, we assumed that reed wood vinegar produced at 500 °C had the highest antibacterial activity against *E. coli*.

In recent years, there has been an urgent need to explore new natural antimicrobial agents considering the problem of increasing bacterial resistance, and it is important to note that reed wood vinegar may be a potential antimicrobial agent and could replace conventional drugs to some extent. The potential use of reed wood vinegar as an alternative antimicrobial agent has been consistently confirmed by research findings, and in China, the agricultural use of reed wood vinegar as a fungicide is regulated by the Certification Bureau of the People’s Republic of China (2012) [39]. In this study, we chose *E. coli* present in the soil obtained from environmental screening, with the aim of investigating the pathway and mechanism of action of reed wood vinegar to inhibit *E. coli* in the living/habitat environments of animals. The aim was to explore the potential impact of reed wood vinegar on sustainable agriculture or environmental restoration. Numerous reviews have commented on the use of wood vinegar as an antimicrobial agent and its bactericidal activity [40,41,42]. de Souza Araújo et al. evaluated the antimicrobial activity of two types of wood vinegar obtained from the slow pyrolysis of *Mimosa tenuiflora* wood and *Eucalyptus urophylla* × *Eucalyptus grandis hybrids* and found that, even at the lowest concentrations of 20% of *E. coli*, *P*. *aeruginosa* and *S*. *aureus* were inhibited by both types of wood vinegar [43]. In the same field of study, wood vinegar from *Mimosa tenuiflora* and *Eucalyptus urograndis* showed significant inhibition of the *S. aureus*, *E. coli*, and *P*. *aeruginosa* strains, showing a circle of inhibition of more than 9 mm [21]. In a previous study, Esquenazi et al. demonstrated that wood vinegar extracted from coconut husk fibres exhibited antimicrobial activity against *S*. *aureus* with antibacterial activity, showing zones of inhibition as high as 13 mm at a concentration of 10 mg/mL [44]. The results of our inhibition circle experiments also showed that the diameter of the inhibition circle of reed wood vinegar with pyrolysis temperature of 300 °C and 500 °C on *E. coli* was between 1 and 1.5 cm, but the reed wood vinegar with a pyrolysis temperature of 700 °C did not show an obvious inhibition circle in the filter paper sheet experiments due to the high concentration of MIC, which cannot indicate that it does not have any inhibitory effect either. In addition, Gama et al. verified the effectiveness of eucalyptus and bamboo wood vinegar as inhibitors of *E. coli* growth by microscopically demonstrating the prominent damage caused by the product to the biofilm [25]. Through SEM observation, we obtained high-definition images of the samples, and we could clearly see the microstructure of *E. coli* on the surface after reed wood vinegar treatment. It can be observed in the electron microscope images that the surface of *E. coli* was regular and full and had a typical bacillus shape [45]. However, morphological changes in the cell walls of all bacteria were observed after exposure to reed wood vinegar compared to the control. The most severely affected microorganisms were the *E. coli* treated with reed wood vinegar at a pyrolysis temperature of 500 °C. The number of bacteria was significantly reduced, their size was reduced, their surfaces were rough, and, in severe cases, the death of bacteria could be observed. This observation confirms the data of the previous sections showing that reed wood vinegar obtained at a 500 °C pyrolysis temperature significantly inhibits the growth viability of *E. coli* and destroys the morphology and biofilm integrity of *E. coli*.

Despite the confirmation of the real potential of reed wood vinegar as an antimicrobial agent and its ability to replace conventional drugs in specific cases, the mechanism of inhibitory action of reed wood vinegar at different pyrolysis temperatures remains a gap. For this reason, we analysed the mechanism of reed wood vinegar inhibition against *E. coli* by RNA-Seq. With the quality of the transcriptome data intact, there was a high degree of clustering of gene structures within the control and MWV groups, and the difference between the two groups was significant, suggesting that reed wood vinegar completely altered the gene structure of *E. coli*. In our case, the results showed that reed wood vinegar affected *E. coli’s* cell wall/membrane generation and its biological functions at multiple levels, as reflected by the following: (1) The DEGs in COG analysis were mainly concentrated in cell wall/membrane/envelope biogenesis; (2) the DEGs in GO enrichment analysis were mainly enriched in the cell membrane fraction; (3) the Ipath diagram demonstrated the obvious effects of reed wood vinegar on the pathways of bacterial membrane function and cellular components in *E. coli* metabolism; (4) among the top 20 differentially expressed genes, *yqiJ*,* yqgA*,* bssR*,* bssS*,** and *ycdT* were correlated with cellular membrane components, and their expression was significantly reduced in reed wood vinegar-treated *E. coli* (*p* < 0.001). Notably, this part of the results corroborates that reed wood vinegar inhibited *E. coli* biofilm structure and production and, thus, exerted an antibacterial effect, which is consistent with a study on the antibiofilm activity of grapefruit seed extracts [46]. In addition, we found that reed wood vinegar was able to influence the production and metabolism of organelles in *E. coli*. DNA synthesis and the assembly of ribosomal large and small subunits are the basis for the composition of *E. coli* [47]. It has been suggested that silencing of RfaH regulates the expression of virulence genes in pathogenic *E. coli* by inhibiting the synthesis of DNA and ribosomes, which in turn inhibits the transcription of nucleoprotein filaments and achieves bacterial inhibition [48]. This suggests us that reed wood vinegar is able to inhibit the growth of *E. coli* fundamentally, which is also an aspect of its bacteriostatic effect. Meanwhile, KEGG results also demonstrated significant enrichment of DEGs in the oxidative phosphorylation-related pathway and significant suppression of genes related to oxidative stress in *E. coli* treated with reed wood vinegar, suggesting that the stimulation of oxidative stress by reed wood vinegar may provide favourable factors for its inhibitory effect.

Finally, we analysed the results of the protein interaction network to propose that the *malE* gene plays an important role in connecting various biological processes in *E. coli*. The *malE* gene is part of *MalEFGK* and is involved in the entry of maltose into the ATP-binding cassette (ABC) transporter complex of *E. coli* [49]. The ABC transporter complex forms a ubiquitous superfamily of membrane complexes that are essential for the active transport of a variety of molecules across the plasma membrane of the *E. coli* [50]. In addition, maltose-binding *malE* within *E. coli* stimulates ATP production; specifically, this process stimulates ATP synthase activity by regressing to an inward-facing conformation and permits the conversion of maltose to ATP [51]. This result also corroborates that the addition of wood vinegar reduced ATP levels in *E. coli*. At the same time, unexpectedly, a large number of studies have shown that *malE* encodes for the generation of Maltose Binding Protein (MBP), which increases the solubility of fusion proteins over-expressed in bacteria, and that many proteins are able to fold into their biologically active conformation once solubilised by fusion with MBP [52,53,54,55,56]. From our results, we can find that the expression of *E. coli malE* decreased significantly after reed wood vinegar treatment, proving that reed wood vinegar may inhibit the protein synthesis process in *E. coli* by suppressing the expression of *malE*, which in turn initiates a series of inhibitory effects. This is something we need to explore further next.

However, the limitations of this paper are mainly the use of a single *E. coli* strain and the potential impact of uncontrolled variables during wood vinegar preparation, as well as the translation of the results of the in vitro studies to an in vivo setting. In the future, we will focus our research on the already-established effects of wood vinegar on different bacteria and further investigate the mechanism of wood vinegar inhibition from the *malE* gene.

## 5. Conclusions

This study confirms that wood vinegar, especially wood vinegar obtained with a 500 °C pyrolysis temperature, is effective as a natural antimicrobial agent against environmental *E. coli*, and provides theoretical references and research ideas for its development in the direction of naturalness and safety. In addition, this study provides insights into the mechanism of action of wood vinegar by proposing the hypothesis that wood vinegar may inhibit the protein synthesis process of *E. coli* by suppressing the expression of *malE*, which in turn initiates a series of antibacterial effects through transcriptome sequencing. Our study lays the foundation for further evaluation of the mechanism of wood vinegar inhibition against environmental *E. coli*.

## Figures and Tables

**Figure 1 biology-13-00912-f001:**
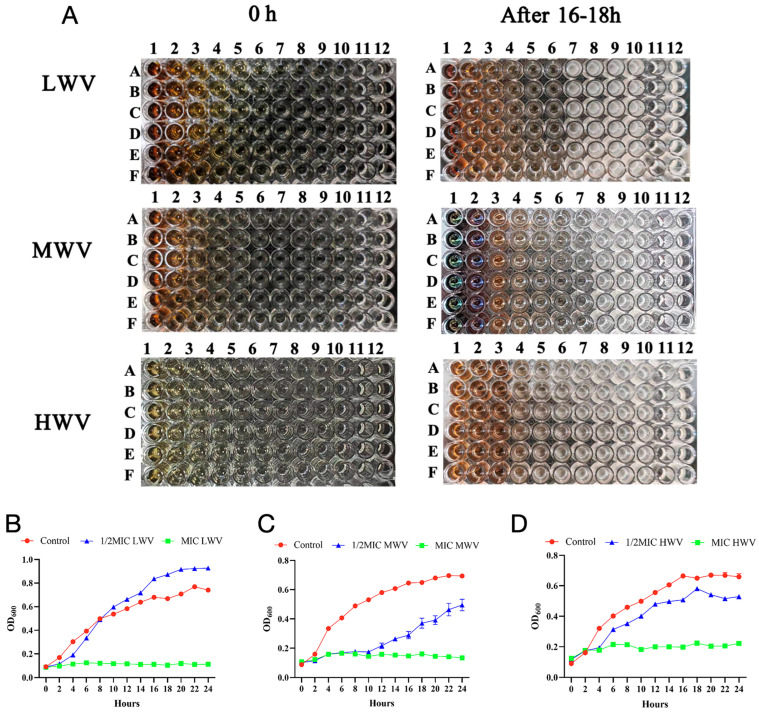
Determination of MIC of wood vinegar and its effect on *E. coli*. (**A**) The MIC of LWV, MWV, and HWV against *E. coli* was determined by nutrient broth dilution method. The drug concentrations in graphs 1–12 are 5.000 mg/mL, 2.500 mg/mL, 1.250 mg/mL, 0.625 mg/mL, 0.313 mg/mL, 0.156 mg/mL, 0.078 mg/mL, 0.039 mg/mL, 0.019 mg/mL, 0.010 mg/mL, 0.005 mg/mL, and 0.003 mg/mL, respectively. (**B**–**D**) Growth curves of *E. coli* without wood vinegar and *E. coli* with the addition of 1/4MIC or 1/2MIC LWV, MWV, and 1/2MIC HWV for 24 h. MIC: minimum inhibitory concentration; LWV: 300 °C wood vinegar; OD: Optical Density; HWV: 700 °C wood vinegar; MWV: 500 °C wood vinegar.

**Figure 2 biology-13-00912-f002:**
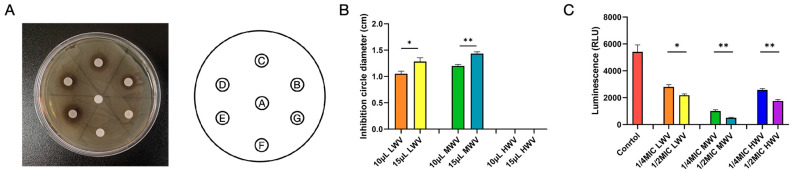
Wood vinegar attenuates *E. coli* growth vigour. (**A**) Paper agar diffusion method was used to detect the inhibitory effect of the same volumes of LWV, MWV, and HWV on the growth of *E. coli*. In the figure on the right, A is saline, B is 10 μL LWV, C is 15 μL LWV, D is 10 μL MWV, E is 15 μL MWV, F is 10 μL LWV, and G is 15 μL LWV. (**B**) Statistical results of the diameter of the circle of inhibition formed after 24 h. (**C**) ATP level assay. Differences between groups were analysed using one-way ANOVA or independent *t*-test: * *p* < 0.05; ** *p* < 0.01.

**Figure 3 biology-13-00912-f003:**
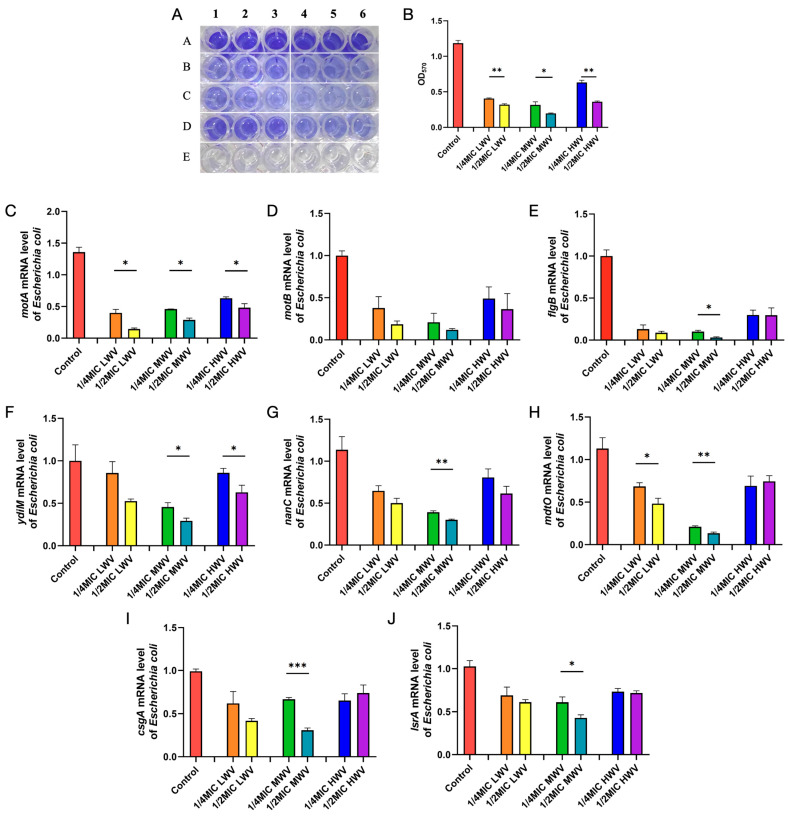
Wood vinegar inhibits *E. coli* biofilm formation and function. (**A**) Quantification of *E. coli* biofilm formation by crystal violet staining. A1-6: controls containing only *E. coli*, B1-3: 1/4MIC LWV, B4-6: 1/2MIC LWV, C1-3: 1/4MIC MWV, C4-6: 1/2MIC MWV, D1-3: 1/4MIC HWV, D4-6: 1/2MIC HWV. E1-6: controls without bacteria. (**B**) Crystalline violet statistics results. The effect of wood vinegar on gene expression in *E. coli* was detected by qPCR. motA (**C**), motB (**D**), flgB (**E**), and csgA (**F**) of *E. coli* biofilm genes; ydiM (**G**), nanC (**H**), and mdtO (**I**) related to cell membrane transport proteins; and lsrA (**J**) related to cellular interactions were found to be inhibited in 1/4MIC and 1/2MIC LWV for LWV, MWV, and HWV. Data are means of three biological replicates; error bars indicate SEM. Differences between groups were analysed using one-way ANOVA or independent *t*-test: * *p* < 0.05; ** *p* < 0.01; *** *p* < 0.001.

**Figure 4 biology-13-00912-f004:**
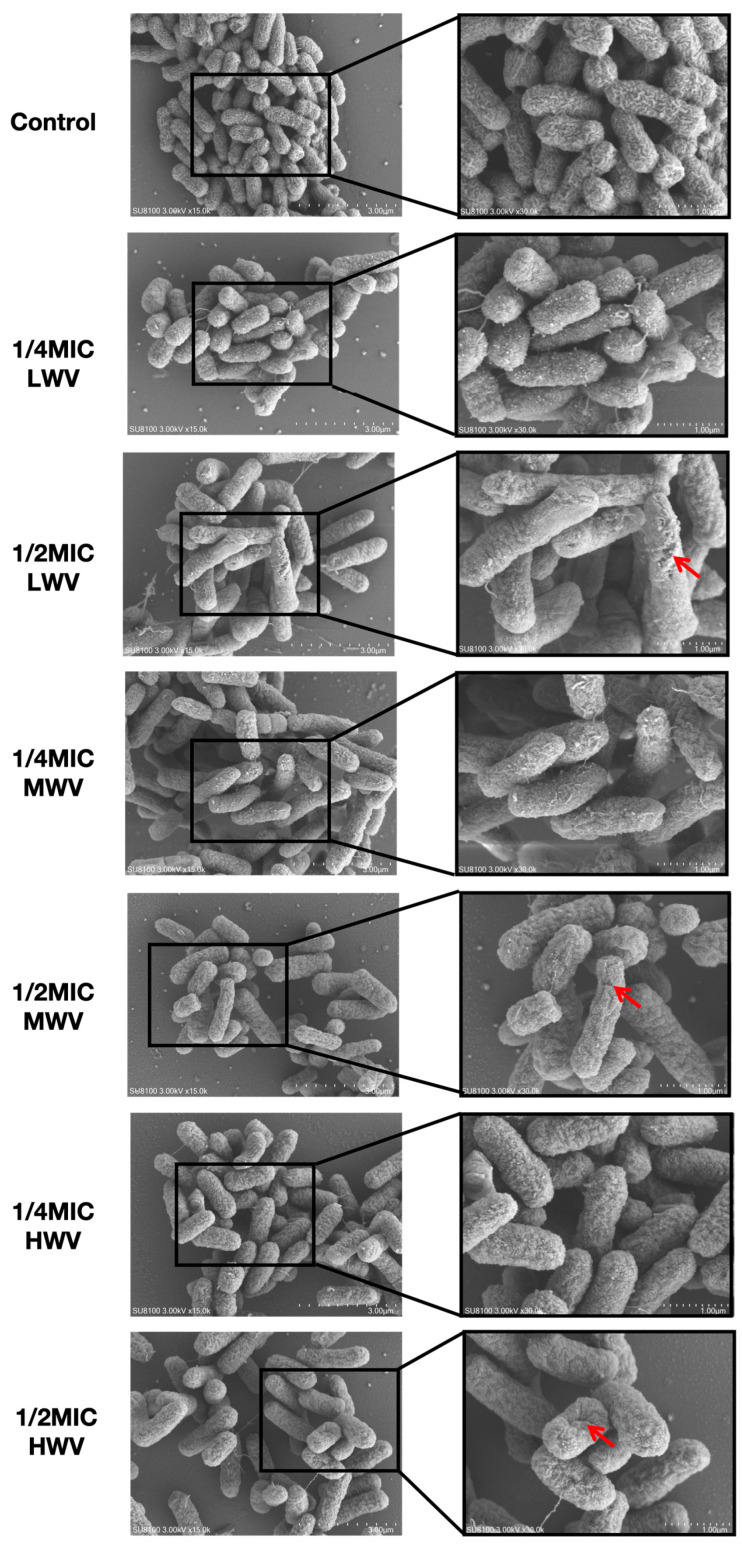
Wood vinegar disruption of *E. coli* biofilm morphology. SEM micrographs of *E. coli* treated and untreated with wood vinegar. The scales in the SEM micrographs represent 3.0 μm (15,000×) and 1.0 μm (30,000×), and the red arrows indicate membrane wrinkling.

**Figure 5 biology-13-00912-f005:**
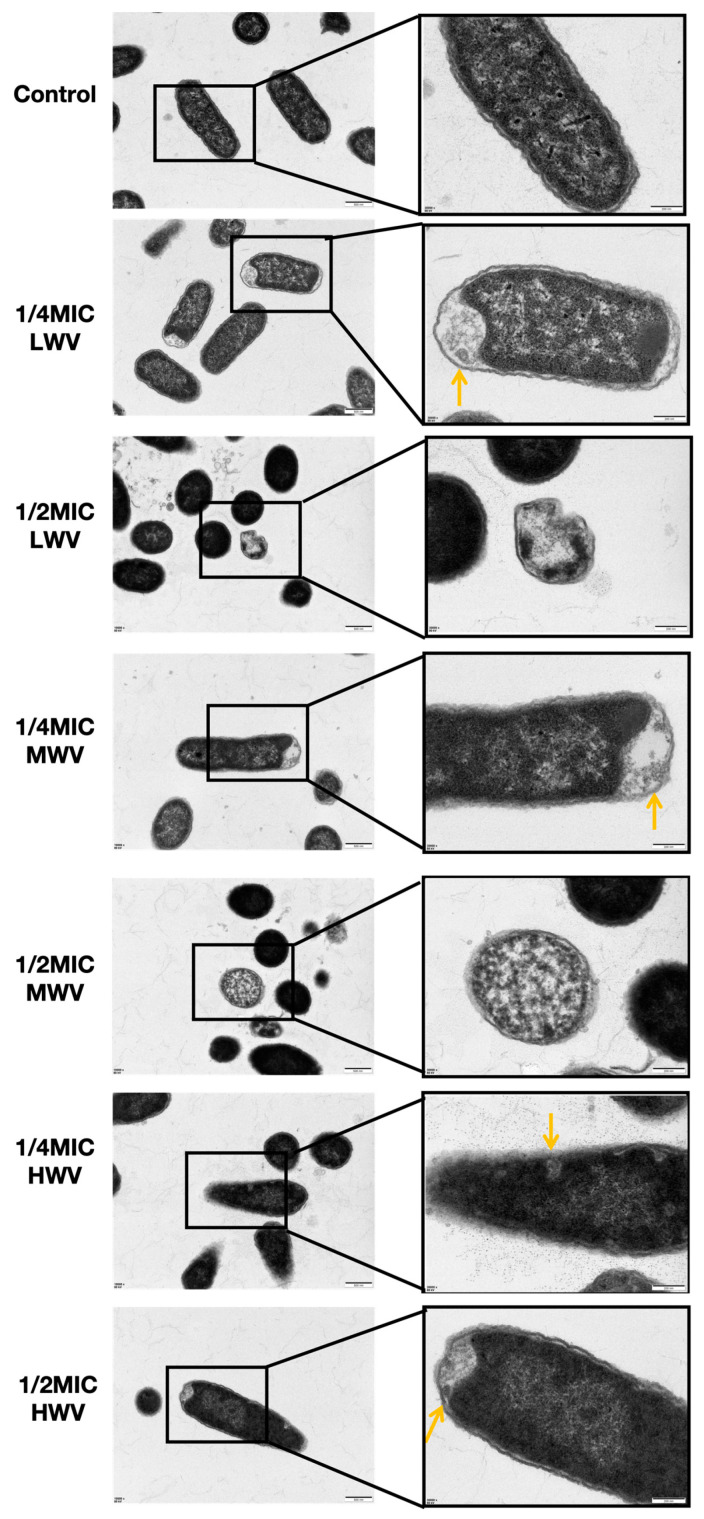
TEM micrographs of *E. coli* treated and untreated with wood vinegar. Each scale of the scales in the transmission electron microscopy micrographs represents 500 nm (10,000×) and 200 nm (30,000×), respectively, and the orange arrows indicate membrane disruption.

**Figure 6 biology-13-00912-f006:**
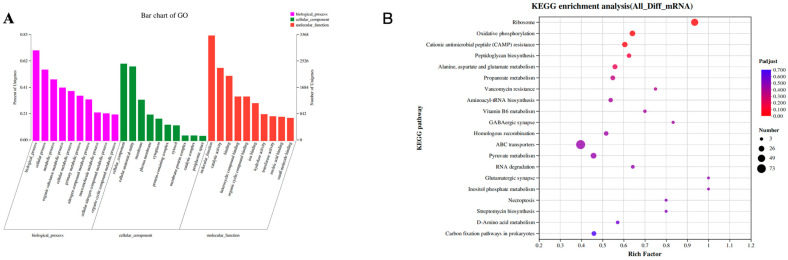
Effect of wood vinegar on the differentially expressed genes of *E. coli*. Transcriptome results for control and MWV groups, including (**A**) Go enrichment analysis; (**B**) KEGG enrichment analysis.

**Figure 7 biology-13-00912-f007:**
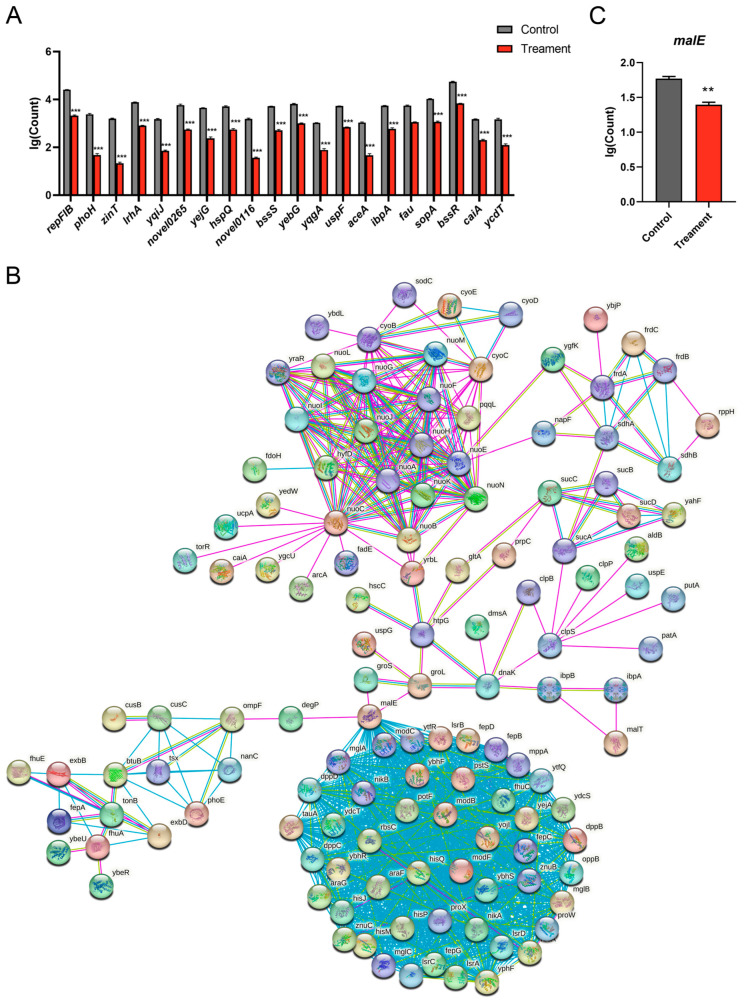
Effect of wood vinegar on gene and protein expression in *E. coli*. (**A**) Top 20 genes with differential expression among 834 DEGs. (**B**) Protein interaction network diagram of proteins edited by 834 DEGs. (**C**) Expression of *malE* genes in transcriptome results. The differences between the experimental values of the two groups were assessed using the Student’s *t*-test with the following statistical significance: ** *p* < 0.01; *** *p* < 0.001.

**Table 1 biology-13-00912-t001:** Primer sequences for housekeeping genes of *E. coli*.

Gene	Primer Direction	Primer Sequence (5′-3′)	Ampliconsize (bp)
*adk*	F	ATTCTGCTTGGCGCTCCGGG	583
	R	CCGTCAACTTTCGCGTATTT	
*fumC*	F	TCACAGGTCGCCAGCGCTTC	806
	R	GTACGCAGCGAAAAAGATTC	
*icd*	F	ATGGAAAGTAAAGTAGTTGTTCCGGCACA	878
	R	GGACGCAGCAGGATCTGTT	
*purA*	F	CGCGCTGATGAAAGAGATGA	816
	R	CATACGGTAAGCCACGCAGA	
*gyrB*	F	TCGGCGACACGGATGACGGC	911
	R	ATCAGGCCTTCACGCGCATC	
*recA*	F	CGCATTCGCTTTACCCTGACC	780
	R	TCGTCGAAATCTACGGACCGGA	
*mdh*	F	ATGAAAGTCGCAGTCCTCGGCGCTGCTGGCGG	932
	R	TTAACGAACTCCTGCCCCAGAGCGATATCTTTCTT	

**Table 2 biology-13-00912-t002:** *E. coli* qPCR primers.

Target Gene	Primer Direction	PCR Primer Sequences (5′-3′)
*motA*	F	TGGAGCACTCTATCAACCCG
	R	CGCCATCAACCGATAAAGCA
*motB*	F	TAGCGGCAAAGTGTTACGTG
	R	GCTTACTGGCTCATTCTGGC
*flgB*	F	ACCTCAACGCAACACATTCC
	R	GCTCATCTGGTATTGCAGGC
*ydiM*	F	GGTCGTCGCTTACCTGTCAT
	R	CGAATGAGTGGAGCGGTAAT
*nanC*	F	AAATGCCGCACTCAATGATGT
	R	GTAACGATAGCGAATGCCAAA
*mdtO*	F	GCCATCCTGCCCACCTTAT
	R	GGCTCCTTGTGCCTGTTCC
*csgA*	F	AGATGTTGGTCAGGGCTCAG
	R	CGTTGTTACCAAAGCCAACC
*lsrA*	F	ACCGAACGCTTGTTTAGTCG
	R	TGTCGTCGGTAGACAGTTCG
*16sRNA*	F	GGTGGCTAAATGCCGTTGTT
	R	TGCGGGGTGTTCATTGTTT

## Data Availability

The datasets generated during and/or analysed during the current study are available in the NCBI repository, https://www.ncbi.nlm.nih.gov/sra/PRJNA1154986, accessed on 1 September 2024.

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
