# Peer review of "Mechanistic Investigation of the Pyrolysis Temperature of Reed Wood Vinegar for Maximising the Antibacterial Activity of Escherichia coli and Its Inhibitory Activity"

_biology, 2024, doi:10.3390/biology13110912_

Round 1
Reviewer 1 Report
Comments and Suggestions for Authors
1. Sharpen the Research Question and Hypotheses:
- Specificity: Clearly define the central research question. Instead of broadly investigating the antibacterial effects of reed wood vinegar, focus on a specific aspect, such as "Determining the optimal pyrolysis temperature for maximizing the antibacterial activity of reed wood vinegar against E. coli and elucidating its underlying mechanism."
- Testable Hypotheses: Formulate clear, testable hypotheses related to the research question. For example: "Hypothesis 1: Reed wood vinegar produced at 500°C will exhibit the highest antimicrobial activity against E. coli. Hypothesis 2: The antimicrobial activity of reed wood vinegar is mediated, at least in part, by downregulating the expression of the malE gene."
2. Enhance the Methods Section for Reproducibility and Rigor:
- Detailed Protocol: Provide a comprehensive, step-by-step description of the wood vinegar preparation method (in supplementary file). Include details about the type of reed used, its pretreatment (e.g., size, drying), pyrolysis parameters (temperature, time, atmosphere), and the purification steps taken to remove impurities (e.g., filtration, activated carbon treatment). Specify the exact method used for the quantification of wood vinegar components (e.g., HPLC, GC-MS).
- Strain Characterization: Provide detailed information about the E. coli strain used in the study (e.g., serotype, origin, relevant genetic information). This is vital for the reproducibility and generalizability of the findings.
- MIC Determination: Specify the exact microbroth dilution method used. Provide information about the positive and negative controls included in the assay.
- RNA Sequencing: Describe precisely the RNA extraction, library preparation, and sequencing protocols used. Specify the sequencing platform used, the read length, and the quality control measures employed (e.g., trimming, adapter removal, quality filtering). Mention the bioinformatic tools and pipelines employed for read mapping, assembly, and differential gene expression analysis (e.g., software versions). State the thresholds applied for determining differentially expressed genes (DEGs) and the statistical methods used to correct for multiple testing.
- Statistical Analysis: For all experimental results (MIC, growth curves, qPCR, biofilm assays), explicitly state the statistical tests used, providing p-values, effect sizes (e.g., Cohen's d), and confidence intervals. Justify the choice of statistical methods used.
3. Strengthen Data Analysis and Interpretation:
- Complete Data Presentation: Include all figures and tables mentioned in the text. Ensure that figures are clearly labeled and include appropriate legends. Provide sufficient numerical data within tables or figures, not only in the text.
- Focus on Key Results: Organize and prioritize the results, highlighting the most important findings. Avoid redundancy and excessive repetition of the same data in different formats.
- Careful Interpretation: Be cautious in interpreting the results. Avoid overstating the conclusions, particularly those related to the proposed mechanism of action. Consider alternative interpretations and limitations of the study. Provide a detailed explanation of the results regarding RNA-Seq analysis, potentially including pathways enriched, functional analysis of the genes, etc.
- Mechanism Elucidation: Provide stronger evidence for the proposed mechanism involving the malE gene. This may involve experiments to validate the effect of wood vinegar on malE expression and the subsequent functional consequences. Consider additional experiments such as gene knockdowns or overexpression experiments to confirm the role of malE.
- Comparative Analysis: Compare the findings with similar studies on the antimicrobial effects of wood vinegar. Discuss the similarities and differences, considering factors such as reed species, pyrolysis conditions, and the target bacteria.
4. Improve the Discussion Section:
- Focus on Mechanism: The discussion should focus on elaborating on the proposed mechanism and address potential alternative explanations for the observed effects. Discuss potential limitations and future research directions related to confirming the specific mechanistic pathway related to malE.
- Contextualization: Place the findings within the broader context of antimicrobial research. Discuss the potential advantages and limitations of using wood vinegar as a natural antimicrobial agent compared to synthetic alternatives. Discuss potential implications for sustainable agriculture or environmental remediation.
- Limitations: Explicitly acknowledge limitations of the study. This includes, but is not limited to: the use of a single E. coli strain, the potential influence of uncontrolled variables during wood vinegar preparation, and limitations in translating in vitrofindings to in vivo settings
Comments on the Quality of English Language
no comments on english
Author Response
Answer to the Reviewers’ Comments
Dear Reviewers of Biology:
Thank you for your letter and for your comments concerning our manuscript entitled “Wood vinegar inhibits bacterial growth and biofilm formation by down-regulating malE gene in Escherichia coli” (biology-3276171). Those comments are all valuable and very helpful for revising and improving our paper. We have studied comments carefully and have made correction which we hope meet with approval. Revised portion are marked in red in the paper. The main corrections in the paper and the responds to the reviewer's comments are as flowing:
Responds to the reviewer's comments:
Reviewer #1
Point 1: Sharpen the Research Question and Hypotheses:
Specificity: Clearly define the central research question. Instead of broadly investigating the antibacterial effects of reed wood vinegar, focus on a specific aspect, such as "Determining the optimal pyrolysis temperature for maximizing the antibacterial activity of reed wood vinegar against E. coli and elucidating its underlying mechanism."
Testable Hypotheses: Formulate clear, testable hypotheses related to the research question. For example: "Hypothesis 1: Reed wood vinegar produced at 500°C will exhibit the highest antimicrobial activity against E. coli. Hypothesis 2: The antimicrobial activity of reed wood vinegar is mediated, at least in part, by downregulating the expression of the malE gene."
Response 1: We thank the reviewers for your comments. We have realigned our research questions and hypotheses to emphasise the exploration of the differences in the inhibitory activity of wood vinegars with different pyrolysis temperatures and their mechanisms. In addition, appropriate assumptions were made in the Discussion section, such as the addition of line 658-660 “Therefore, we assumed that reed wood vinegar produced at 500°C had the highest antibacterial activity against E. coli.”
Pleases see the Introduction and Discussion section.
Point 2: Enhance the Methods Section for Reproducibility and Rigor:
Detailed Protocol: Provide a comprehensive, step-by-step description of the wood vinegar preparation method (in supplementary file). Include details about the type of reed used, its pretreatment (e.g., size, drying), pyrolysis parameters (temperature, time, atmosphere), and the purification steps taken to remove impurities (e.g., filtration, activated carbon treatment). Specify the exact method used for the quantification of wood vinegar components (e.g., HPLC, GC-MS).
Strain Characterization: Provide detailed information about the E. coli strain used in the study (e.g., serotype, origin, relevant genetic information). This is vital for the reproducibility and generalizability of the findings.
MIC Determination: Specify the exact microbroth dilution method used. Provide information about the positive and negative controls included in the assay.
RNA Sequencing: Describe precisely the RNA extraction, library preparation, and sequencing protocols used. Specify the sequencing platform used, the read length, and the quality control measures employed (e.g., trimming, adapter removal, quality filtering). Mention the bioinformatic tools and pipelines employed for read mapping, assembly, and differential gene expression analysis (e.g., software versions). State the thresholds applied for determining differentially expressed genes (DEGs) and the statistical methods used to correct for multiple testing.
Statistical Analysis: For all experimental results (MIC, growth curves, qPCR, biofilm assays), explicitly state the statistical tests used, providing p-values, effect sizes (e.g., Cohen's d), and confidence intervals. Justify the choice of statistical methods used.
Response 2: We are very grateful about the reviewer's suggestion. We have provided a comprehensive, step-by-step description of the methodology for the preparation of wood vinegar in a supplementary document.
About Strain Characterisation, the E. coli in this study used for experiments in this study were obtained by collecting, isolating, and culturing from the soil around the feces of Milu in the Beijing Milu Park. The single E. coli was obtained by plate streaking method in MacConkey isolation medium for 24h followed by monoclonal culture for 24h. The bacterial type (ST) was later identified by the MLST method with the NCBI database as 101.
About MIC Determination, the bacterial OD600 was diluted to 0.5 (approximately 1.0×108 CFUs/mL) using spectrophotometer (Shanghai Jinghua, China) and was prepared by diluting it 100-fold with BHI broth medium. Then, the reeds wood vinegar at different pyrolysis temperatures (LWV, MWV and HWV) was diluted by 2 times to the 10th well in a 96-well plate with 100 μL of Brain-Heart Infusion Broth (BHI) medium added. In addition, the 11th column was a negative control containing only BHI medium and the 12th column was a positive control containing the bacterial solution to be tested, respectively. After dilution, 100 µL of the corresponding microbial inoculum was added to each microtiter well.
We also supplemented the RNA sequencing and statistical analyses section with a description of the RNA extraction, library preparation, and sequencing protocols, as well as a clear description of the statistical tests used and the rationale for choosing the statistical methods used.
Pleases see the Materials and Methods section.
Point 3: Strengthen Data Analysis and Interpretation:
Complete Data Presentation: Include all figures and tables mentioned in the text. Ensure that figures are clearly labeled and include appropriate legends. Provide sufficient numerical data within tables or figures, not only in the text.
Focus on Key Results: Organize and prioritize the results, highlighting the most important findings. Avoid redundancy and excessive repetition of the same data in different formats.
Careful Interpretation: Be cautious in interpreting the results. Avoid overstating the conclusions, particularly those related to the proposed mechanism of action. Consider alternative interpretations and limitations of the study. Provide a detailed explanation of the results regarding RNA-Seq analysis, potentially including pathways enriched, functional analysis of the genes, etc.
Mechanism Elucidation: Provide stronger evidence for the proposed mechanism involving the malE gene. This may involve experiments to validate the effect of wood vinegar on malE expression and the subsequent functional consequences. Consider additional experiments such as gene knockdowns or overexpression experiments to confirm the role of malE.
Comparative Analysis: Compare the findings with similar studies on the antimicrobial effects of wood vinegar. Discuss the similarities and differences, considering factors such as reed species, pyrolysis conditions, and the target bacteria.
Response 3: We have made correction according to the reviewer's comments. We have made a new layout of Figure 4 and worked to address the unclear length scale. In addition, we have inserted the supplementary chart in the corresponding position in the Results section.
In addition, the results of the RNA-Seq analyses are explained in detail in the results section, for example “Moreover, we performed lineage analysis of the GO pathway for structural relationships, and from Figure S4 E, we were able to clearly find that the assembly of ribosomal large and small subunits by MWV is the underlying cause of the effect on the structure of E. coli, which further affects the protein-RNA complex assembly. In addition, the assembly of ribosomal large and small subunits affects bacterial membrane function and cellular composition . KEGG pathway enrichment aggregation revealed significant differences in the expression of genes primarily involved in ribosome, oxidative phosphorylation and cationic antimicrobial peptide (CAMP) resistance (Figure 6 B).”
We propose that the malE gene may play a role in the mechanism of wood vinegar suppression, with a modified title. And in the discussion section we have analysed wood vinegars from different sources, with different pyrolysis temperatures and inhibition of different bacteria, whereas there are fewer studies related to reed wood vinegar and therefore it is not discussed.
Pleases see the Results section.
Point 4: Improve the Discussion Section:
Focus on Mechanism: The discussion should focus on elaborating on the proposed mechanism and address potential alternative explanations for the observed effects. Discuss potential limitations and future research directions related to confirming the specific mechanistic pathway related to malE.
Contextualization: Place the findings within the broader context of antimicrobial research. Discuss the potential advantages and limitations of using wood vinegar as a natural antimicrobial agent compared to synthetic alternatives. Discuss potential implications for sustainable agriculture or environmental remediation.
Limitations: Explicitly acknowledge limitations of the study. This includes, but is not limited to: the use of a single E. coli strain, the potential influence of uncontrolled variables during wood vinegar preparation, and limitations in translating in vitro findings to in vivo settings
Response 4: Thank you for your suggestions for these issues. We discussed the possible role of the malE gene in the mechanism of wood vinegar inhibition and modified the title.
And we have situated the findings in the broader context of antimicrobial research in our discussion and added “In this study, we chose E. coli present in the soil obtained from environmental screening, with the aim of investigating the pathway and mechanism of action of reeds wood vinegar to inhibit E. coli in the animal living/habitat environment. The aim was to explore the potential impact of reed wood vinegar on sustainable agriculture or environmental restoration.” this sentence to the article.
In addition, we describe the limitations of the article and add to it as follows: “However, the limitations of this paper are mainly the use of a single E. coli strain and the potential impact of uncontrolled variables during wood vinegar preparation, as well as the translation of the results of the in vitro studies to the in vivo setting. In the future, we will focus our research on the already effects of wood vinegar on different bacteria and further investigate the mechanism of wood vinegar inhibition from the malE gene”
Pleases see the Discussion section.
These are my answers to the questions posed by reviewer 1. Special thanks to you for your good comments
We tried our best to improve the manuscript and made some changes in th emanuscript. These changes will not influence the content and framework of the paper. And here we did not list the changes but marked in revised paper.
We appreciate for Editors/Reviewers' warm work earnestly, and hope that the correction will meet with approval.
Once again, thank you very much for your comments and suggestions.
Kind regards,
Bai Bing
Corresponding author:
Name: Qingyun Guo, Jingjing Yao
E-mail: guoqingyun1987@126.com (Q.G.); yaojing1989_lucky@163.com (J.Y.)

Reviewer 2 Report
Comments and Suggestions for Authors
In this study, authors found that 1/2MIC 500°C wood vinegar had the most prominent bacteria inhibitory activity. qPCR results showed that reed wood vinegar was able to significantly inhibit the expression of E. coli biofilm and genes related to the cell membrane transporter proteins. RNA-Seq showed the multifaceted antimicrobial effects of wood vinegar and demonstrated that the inhibitory effect of wood vinegar on E. coli was mainly realized through the inhibition of the expression of malE. This work is well done and its main results look interesting by comprising several technological aspects. It can be considered to be published after running moderate revision.
Questions:
1. Title, The results presented in the article are insufficient to demonstrate that wood vinegar inhibits bacterial growth and biofilm formation by down-regulating malE gene in Escherichia coli. Authors should revise the title.
2. Line 11-21, “Escherichia coli” and “E. coli” should be in italics.
3. The strain isolated from the soil around the feces of Milu in the Beijing Milu Park should be strictly identified to confirm that it is E. coli, at least by 16S rDNA sequencing.
4. Section 2.7, It is necessary to explain the rationale for selecting nine cell membrane genes for qPCR.
5. Figure 1, It is advised to add the concentration information for samples in the 1-12 columns.
6. Figure 4, The length scale is very unclear. Please check it.
7. The malE gene is not in the top 20 genes from the 834 genes with the most 530 obvious differences. And they play important roles in cell membrane components, oxidative stress, DNA binding and repair. Whereas, authors believes that the role of malE is more significant. More comprehensive references and data support need to be supplemented.
Author Response
Answer to the Reviewers’ Comments
Dear Reviewers of Biology:
Thank you for your letter and for your comments concerning our manuscript entitled “Wood vinegar inhibits bacterial growth and biofilm formation by down-regulating malE gene in Escherichia coli” (biology-3276171). Those comments are all valuable and very helpful for revising and improving our paper. We have studied comments carefully and have made correction which we hope meet with approval. Revised portion are marked in red in the paper. The main corrections in the paper and the responds to the reviewer's comments are as flowing:
Responds to the reviewer's comments:
Reviewer #2
Point 1: Title, The results presented in the article are insufficient to demonstrate that wood vinegar inhibits bacterial growth and biofilm formation by down-regulating malE gene in Escherichia coli. Authors should revise the title.
Response 1: We thank the reviewer for pointing out that thing. We have changed the title to”Mechanistic investigation of the pyrolysis temperature of reed wood vinegar for maximising the antibacterial activity of Escherichia coli and its inhibitory activity”.
Pleases see the Tile.
Point 2: Line 11-21, “Escherichia coli” and “E. coli” should be in italics.
Response 2: We thank the reviewer for pointing out that thing. We carefully checked the “Escherichia coli” and “E. coli” throughout the Simple Summary section and changed their formatting to italics.
Pleases see line 12-22.
Point 3: The strain isolated from the soil around the feces of Milu in the Beijing Milu Park should be strictly identified to confirm that it is E. coli, at least by 16S rDNA sequencing.
Response 3: Considering the reviewer's suggestion, we need to explain the strain isolation and identification methods. In section 2.1 of Materials and Methods it is written that we used MacConkey's medium to isolate the red strain. After that we also detected 7 housekeeping genes of E. coli and identified its E. coli ST type according to NCBI's MLST method in the 2.2 part of Materials and Methods. Therefore, based on the results of the above two experiments we were able to prove that the strain we isolated was E. coli.
Pleases see 2.1 and 2.2 section of Materials and Methods.
Point 4: Section 2.7, It is necessary to explain the rationale for selecting nine cell membrane genes for qPCR.
Response 4: We thank the reviewer for pointing out that thing. We have added references to the selection of nine cell membrane genes for qPCR in Section 2.7.
Pleases see line 248 of Materials and Methods.
Point 5: Figure 1, It is advised to add the concentration information for samples in the 1-12 columns.
Response 5: Thank you for your suggestions for the figure. We have added a description of the drug concentration in each well in the annotation below the figure. Specifically, “the drug concentrations in graphs 1-12 are 5.000 mg/ml, 2.500 mg/ml, 1.250 mg/ml, 0.625 mg/ml, 0.313 mg/ml, 0.156 mg/ml, 0.078 mg/ml, 0.039 mg/ml, 0.019 mg/ml, 0.010 mg/ml, 0.005 mg/ml, 0.003 mg/ml respectively.”
Pleases see line 442-445.
Point 6: Figure 4, The length scale is very unclear. Please check it.
Response 6: We are sorry for the inconvenience. We have made a new layout of Figure 4 and worked to address the unclear length scale.
Pleases see Figure 4.
Point 7: The malE gene is not in the top 20 genes from the 834 genes with the most 530 obvious differences. And they play important roles in cell membrane components, oxidative stress, DNA binding and repair. Whereas, authors believes that the role of malE is more significant. More comprehensive references and data support need to be supplemented.
Response 7: We thank the reviewers for your comments. In order to better articulate the content of our study, we have changed the title to”Mechanistic investigation of the pyrolysis temperature of reed wood vinegar for maximising the antibacterial activity of Escherichia coli and its inhibitory activity”.
Pleases see the Tile.
These are my answers to the questions posed by reviewer 2. Special thanks to you for your good comments
We tried our best to improve the manuscript and made some changes in th emanuscript. These changes will not influence the content and framework of the paper. And here we did not list the changes but marked in revised paper.
We appreciate for Editors/Reviewers' warm work earnestly, and hope that the correction will meet with approval.
Once again, thank you very much for your comments and suggestions.
Kind regards,
Bai Bing
Corresponding author:
Name: Qingyun Guo, Jingjing Yao
E-mail: guoqingyun1987@126.com (Q.G.); yaojing1989_lucky@163.com (J.Y.)

Reviewer 3 Report
Comments and Suggestions for Authors
The manuscript showed the novelty of reed wood vinegar for a natural antimicrobial with clear mechanism of bacterial inhibition
1. At first glance, many gene names and species names are not italicized throughout the manuscript. Please check.
2. In line 16, “in the environment” à “in vitro”.
3. The statement in lines 20-21 is not necessary because “wood vinegar” has been proven as “a natural antimicrobial agent” in line 14.
4. Check the coma in line 25.
5. Please define “malE” gene in line 33. This could make the abstract smoother.
6. Rewrite the conclusion in the abstract. It is acceptable, but it repeats what has been mentioned at the background of the abstract.
7. Name the Figure in line 39, e.g. “Graphical Abstract”.
8. Please improve the resolution of the Figure of “RNA-Seq of E. coli”. The quality is too poor to read.
9. Although the part in lines 42-56 is good and adequate to introduce 56, try to make it a bit shorter and more concise.
10. Please cite the statement in lines 65-68 with proper literatures.
11. Latinate words should be italicized.
12. In line 83, “Study” à “A study”.
13. In line 87, remove “)”.
14. Cite in line 100.
15. Please reformat the Table 1.
16. Many abbreviated terms have not been defined at first mention.
17. What were the source of primers in Table 2.
18. Figures and Tables should appear right after their first mention.
19. Results should not contain references.
20. The conclusion should be extended with the contributions to the current field and the further applications or investigations.
Author Response
Answer to the Reviewers’ Comments
Dear Reviewers of Biology:
Thank you for your letter and for your comments concerning our manuscript entitled “Wood vinegar inhibits bacterial growth and biofilm formation by down-regulating malE gene in Escherichia coli” (biology-3276171). Those comments are all valuable and very helpful for revising and improving our paper. We have studied comments carefully and have made correction which we hope meet with approval. Revised portion are marked in red in the paper. The main corrections in the paper and the responds to the reviewer's comments are as flowing:
Responds to the reviewer's comments:
Reviewer #3
Point 1: At first glance, many gene names and species names are not italicized throughout the manuscript. Please check.
Response 1: We thank the reviewer for pointing out that thing. We carefully checked the gene and bacterial names throughout the text, especially in the “Simple Summary” section, and changed their formatting to italics.
Pleases see the full text.
Point 2: In line 16, “in the environment” à “in vitro”.
Response 2: We have made correction according to the reviewer's comments. We have amended “in the environment” to “in vitro” in the “Simple Summary” section.
Please see line 17.
Point 3: The statement in lines 20-21 is not necessary because “wood vinegar” has been proven as “a natural antimicrobial agent” in line 14.
Response 3: We thank the reviewer for pointing out that thing. We have changed “This provides a theoretical basis for the potential of wood vinegar as a natural antimicrobial agent.” to “This provides a theoretical basis for the mechanism of wood vinegar as a natural antimicrobial agent.”
Please see line 23.
Point 4: Check the coma in line 25.
Response 4: We thank the reviewers for your comments. We have replaced “、” with “,”.
Please see line 27.
Point 5: Please define “malE” gene in line 33. This could make the abstract smoother.
Response 5: We are very grateful about the reviewer's suggestion. We defined the malE gene and added the definition “which is an ATP-binding cassette (ABC) transporter complex of E. coli.”
Please see line 35.
Point 6: Rewrite the conclusion in the abstract. It is acceptable, but it repeats what has been mentioned at the background of the abstract.
Response 6: Thank you for your suggestions for the abstract. We have changed “In conclusion, our study provides an effective method provides a theoretical basis for the potential of reed wood vinegar as a natural antimicrobial agent and its mechanism of bacterial inhibition.” to “In conclusion, our study provides an effective method provides a theoretical basis for the mechanism of reed wood vinegar as a natural antimicrobial agent and its pathway of bacterial inhibition.”
Please see line 37-39 of the abstract.
Point 7: Name the Figure in line 39, e.g. “Graphical Abstract”.
Response 7: We have added “Graphical Abstract” at the appropriate place in the manuscript.
Please see line 42.
Point 8: Please improve the resolution of the Figure of “RNA-Seq of E. coli”. The quality is too poor to read.
Response 8: We are sorry for the inconvenience. We have improved the resolution of the “RNA-Seq of E. coli” figure in the Graphical Abstract and re-uploaded it in the revised manuscript.
Please see Graphical Abstract.
Point 9: Although the part in lines 42-56 is good and adequate to introduce 56, try to make it a bit shorter and more concise.
Response 9: We thank the reviewers for your comments. We have changed the content before line 56 as follows: “As the main wetland species in China, reeds play a very important role in water purification, regional climate regulation, and wetland species protection [1]. With the massive growth of reeds, the withered reeds will decompose into water bodies after withering, causing eutrophication of wetland ecosystems and the production of black and odorous water bodies [2]. Currently, in order to dispose of excess reeds around wetlands, the main routes are through landfill [3], burning [4, 5] or pyrolysis. Pyrolysis is a thermochemical process for the production of biochar as a sustainable biomass conversion and waste management method [6]. In order to address the excess vapour or fumes generated during pyrolysis [7], plant recovery and proper disposal of pyrolysis fluids is one way to turn the situation around, resulting in additional economic benefits, significant reductions in greenhouse gas emissions, and high-quality products with a variety of applications [8].”
Please see the Introduction section.
Point 10: Please cite the statement in lines 65-68 with proper literatures.
Response 10: We thank the reviewers for your comments. We have added the appropriate documentation when quoting the statement in lines 65-68.
Please see line 70 of the Introduction.
Point 11: Latinate words should be italicized.
Response 11: We thank the reviewer for pointing out that thing. We carefully checked the Latinate words throughout the text and changed their formatting to italics.
Pleases see the full text.
Point 12: In line 83, “Study” à “A study”.
Response 12: We thank the reviewer for pointing out that thing. We have changed “Study” to “A study”.
Pleases see line 86 of the Introduction.
Point 13: In line 87, remove “)”.
Response 13: We thank the reviewer for pointing out that thing. We have removed “)”.
Pleases see line 90 of the Introduction.
Point 14: Cite in line 100.
Response 14: We thank the reviewers for your comments. We have added a reference to the content of lines 102-104.
Pleases see line 105 of the Introduction .
Point 15: Please reformat the Table 1.
Response 15: We have made correction according to the reviewer's comments. We have reformatted Table 1 and put “Primer direction” in a separate column.
Pleases see line 162.
Point 16: Many abbreviated terms have not been defined at first mention.
Response 16: Thank you for your suggestions for this issue. We checked the full text, defined acronyms when they were first mentioned, and supplemented the glossary of acronyms.
Pleases see the full text.
Point 17: What were the source of primers in Table 2.
Response 17: We have made correction according to the reviewer's comments. Primers for qPCR were obtained by querying the gene database at NCBI (https://www.ncbi.nlm.nih.gov/) and synthesised by Bioengineering Biotechnology (Shanghai) Co Ltd (Table 2).
Pleases see line 258 of the Materials and Methods.
Point 18: Figures and Tables should appear right after their first mention.
Response 18: We are sorry for the inconvenience. We have inserted the supplementary chart in the corresponding position in the Results section.
Pleases see the Results section.
Point 19: Results should not contain references.
Response 19: We thank the reviewers for your comments. We have removed the references from the results.
Pleases see the Results section.
Point 20: The conclusion should be extended with the contributions to the current field and the further applications or investigations.
Response 20: Thank you for your suggestions for this section. We have changed the conclusion to “This study confirms that wood vinegar, especially wood vinegar obtained from 500°C pyrolysis temperature, is effective as a natural antimicrobial agent against environmental E. coli, and provides theoretical references and research ideas for its development in the direction of naturalness and safety. In addition, this study provides insights into the mechanism of action of wood vinegar by proposing the hypothesis that wood vinegar may inhibit the protein synthesis process of E. coli by suppressing the expression of malE, which in turn initiates a series of antibacterial effects through transcriptome sequencing. Our study lays the foundation for further evaluation of the mechanism of wood vinegar inhibition against environmental E. coli”
Pleases see the Conclusion section.
These are my answers to the questions posed by reviewer 3. Special thanks to you for your good comments.
We tried our best to improve the manuscript and made some changes in th emanuscript. These changes will not influence the content and framework of the paper. And here we did not list the changes but marked in revised paper.
We appreciate for Editors/Reviewers' warm work earnestly, and hope that the correction will meet with approval.
Once again, thank you very much for your comments and suggestions.
Kind regards,
Bai Bing
Corresponding author:
Name: Qingyun Guo, Jingjing Yao
E-mail: guoqingyun1987@126.com (Q.G.); yaojing1989_lucky@163.com (J.Y.)

Round 2
Reviewer 2 Report
Comments and Suggestions for Authors
Authors has answered all questions and the manuscript could be accepted.